# The Influence of Depressive Mood on Mortality in Elderly with Different Health Status: Evidence from the Taiwan Longitudinal Study on Aging (TLSA)

**DOI:** 10.3390/ijerph19116922

**Published:** 2022-06-06

**Authors:** Shen-Ju Tsai, Yu-Han Hsiao, Miao-Yu Liao, Meng-Chih Lee

**Affiliations:** 1Department of Family Medicine, Everan Hospital, Taichung 411001, Taiwan; drrudy1985@gmail.com; 2Institute of Public Health, National Yang Ming Chiao Tung University, Taipei 112304, Taiwan; 3Department of Family Medicine, Taichung Hospital, Ministry of Health and Welfare, Taichung 403301, Taiwan; phoebe01026@gmail.com (Y.-H.H.); liaomiaoyu@gmail.com (M.-Y.L.); 4College of Management, Chaoyang University of Technology, Taichung 413310, Taiwan; 5Department of Public Health, Chung Shan Medical University, Taichung 402306, Taiwan; 6Institute of Population Health Sciences, National Health Research Institutes, Miaoli 350401, Taiwan; 7Institute of Medicine, Chung Shan Medical University, Taichung 402306, Taiwan

**Keywords:** depression, mortality, self-rated health, CES-D, TLSA

## Abstract

Depression and related syndromes are well identified in older adults. Depression has been reported to increase the incidence of a multitude of somatic disorders. In older adults, the severity of depression is associated with higher mortality rates. The aim of the study is to examine whether the effect of depression screening on mortality is different between individuals with different physical health status. In order to meet this aim, we will first reprove the relationship between depression and mortality rate, and then we will set a subgroup analysis by using self-reported health (SRH) status. Our data source, Taiwan Longitudinal Study on Aging (TLSA), is a population-based prospective cohort study that was initiated by the Health Promotion Administration, Ministry of Health and Welfare, Taiwan. The depression risk was evaluated by 10-items Center for Epidemiologic Studies Depression (CES-D-10), we set 3 CES-D-10 cutting points (5, 10, and 12) and cut our subjects into four groups. Taking mortality as an end point, we use the Taiwan National Death Registry (TNDR) record from 1999 to 2012. Self-rated health (SRH) was taken as an effect modifier between depression and mortality in the elderly group, and stratification took place into three groups (good, fair, poor). The case numbers of 4 CES-D-10 groups were 2253, 939, 285 and 522, respectively. After dividing into 4 CES-D-10 groups, the mortality prevalence rose as the CES-D-10 level grew (40.7%, 47.82%, 54.39% and 67.62%, respectively). In the subgroup analysis, although the *p*-value of log-rank test showed <0.05 in three groups, as the SRH got worse the Hazard Ratio became more significant (*p* = 0.122, 0.033, <0.001, respectively). Kaplan–Meier (K-M) survival estimates for different CES-D groups in SRH were poor, and we can see the curves representing second and third CES-D group going almost together, which may suggest the cutting point of CES-D-10 in predicting depression risk should be adjusted in the relatively unhealthy elderly. The importance of the relationship between depression and mortality is re-emphasized in our study. Moreover, through joining SRH in our analysis, we can conclude that in self-rated poor health any sign of depression may lead to a rise in mortality. Therefore, we should pay attention to the old age group’s psychological status, and remember that depressive mood should be scrutinized more carefully in the elderly who feel themselves to be unhealthy.

## 1. Introduction

Depression and its related syndromes are well identified in older adults, although the mechanism is not clear [1,2]. The prevalence for depressive symptoms is around 10% to 15% in older adults [3]. Depression has been reported to increase the incidence of a multitude of somatic disorders [4], but research also suggests an opposing view regarding causality [5].

Depression risk is often measured as a surrogate of depression in older adults [6]. In order to identify elders with depressive symptoms, we used validated scales such as the Center for Epidemiologic Studies Depression (CES-D) from Radloff in 1977, which has proved similar when comparing with the Geriatric Depression Scale (GDS) in identifying cases with depressive disorder in diagnostic interview [7,8].

Depression is associated with increased mortality rates [9,10,11]. In older adults, the severity of depression is associated with higher mortality rates [12,13]. Increasing somatic disease burden, drug adverse effects, unhealthy lifestyle/social determinants, and refusing health care can be important [14]. Many meta-analyses have been carried out in order to confirm the association between depression and increasing mortality rates, but with low quality of the evidence and without age specification [15,16]. Furthermore, in different types of physical disorder, depression is considered to be an important prognosis predicting factor. Some types of depression have also been associated with an increased incidence of metabolic syndrome [17]. In following studies, depression was also found to be connected with increasing mortality in stroke [18], myocardial infarction [19], type II diabetes [20], and even frailty [21]. A recent meta-analysis demonstrated the association of late-life depression with all cause and cardiovascular mortality among community-dwelling older adults, which showed around a 30% increase [22]. In the geriatric area, some risk factors of mortality, such as unhealthy lifestyle characteristics and multiple drug prescriptions, have also been demonstrated to be important in the depressive older group [23]. All evidence points out the close correlation between physical health, psychological health and mortality.

We can expect that mortality risk may increase in older people with simultaneous depression and somatic disorder, which allows us to examine whether the effect of depression screening on mortality is different between individuals with different physical health status. As our first objective, we are going to reprove the relationship between depression and mortality rate through a longitudinal study, the Taiwan Longitudinal Study on Aging (TLSA), over 12 years of observation (1999–2012). As our second objective, we will set a subgroup analysis by using the self-reported health (SRH) status, in order to see how different health status affects the relationship between depressive mood and mortality.

## 2. Methods

### 2.1. Data Source

The TLSA is a national, population-based longitudinal study that was initiated by the Health Promotion Administration, Ministry of Health and Welfare, Taiwan. This survey was first conducted in 1989, and 4049 older adult residents were included. A three-stage systematic random sampling design was used for the sample selection [24]. Data were collected by a face-to-face interview conducted by well-trained interviewers with questionnaire completion rate up to 90%. If participants had significant cognitive impairments, the answer would be provided by their proxies. The individuals were followed up every 3–4 years. Another 2462 population samples were selected in 1996 to maintain representativeness of the younger age cohort and to extend the representativeness of the sample to the population aged 50 and above. In this study, we used data collected in 1999 (Wave IV). All of the participants provided informed consent, and their anonymity was preserved. The details and design of the TLSA have been described elsewhere [24,25,26].

### 2.2. Subjects, Study Design, Setting, and Ethical Aspects

Under the setting of longitudinal study, we analyzed 4440 sample subjects collected in the 4th wave of the TLSA survey in 1999, taken as a starting point, consisting of 2310 subjects included in 1989 and 2130 subjects in 1996 (another 1739 subjects from 1989 and 152 subjects from 1996 have been lost due to expired data, migration or unknown reasons). During analysis, we excluded 340 subjects due to non-completion of the CES-D form, 53 subjects due to no specific interview date, and 48 subjects due to lost follow up without any explicit reason. Only 3999 subjects were included in our data analysis (Figure 1). The time period started from the date of the 4th wave of the TLSA interview, from 1999 to 31 December 2012 if the case was alive. If the case had expired, we took this as an event, and used the date of death as the endpoint.

### 2.3. Research Variables

#### 2.3.1. Independent Variable: Depression Risk by CES-D-10 Point

Depression risk was evaluated with the 10-item CES-D(CES-D-10) [7,8]. We rated depressive symptoms “during the past week” with the CES-D-10 rating scale. The 10-items CES-D (CES-D-10), with a score range from zero to thirty, has the same sensitivity and specificity as the 20-item CES-D (CES-D-20) scale [27] and performs well in older Chinese [28]. Although the credibility of CES-D-10 was sufficient, the cutting point of CES-D-10 was fractured. The most commonly used cutting point for CES-D-10 is above 10 [29,30,31], but some more rigorous study have stated that, according to the CES-D-20 (score range 0–60), the cutting point for CES-D-10 should be above 13 since the original CES-D-20 takes a score of above 25 to rate depressive symptoms [32]. However, according to research evaluating the CES-D in Iranian elderly [33], when CES-D-10 was above 5, it already met the criteria as a screening tool for the elderly according to the ROC curve evaluation. All these cutting points seem to provide some degree of detection of varying sensitivities to depression.

The range of CES-D-10 was from 0 to 30. According to the debate about the cutting point of CES-D-10 in our introduction, we will use 3 cutting points (5, 10, 13) and cut the CES-D-10 score into 4 groups in order to see the change in correlation between depression and mortality: no depression (ND), depression potential (DP), mild depression (MD), and depression disorder (DD). The ND group (CES-D-10 < 5) is considered to be the baseline group with minimal depression possibility. The DP group (5 ≦ CES-D-10 < 10) is considered to be the group in a relatively depressive mood, but may not precisely fit the diagnosis of depression [33]. The MD group (10 ≦ CES-D-10 < 13) us3s the most common cut-off point to define depressive symptoms [29,30,31]. The DD group (13 ≦ CES-D-10) uses the most rigorous criteria, and is considered to be very likely to have depression disorder [32].

#### 2.3.2. Dependent Variable: Mortality

We examined the effect of depressive mood or depression on the mortality of older adults in Taiwan. Taking mortality as our major variable, the participants’ national identification number was used to link the data from TLSA and Taiwan National Death Registry (TNDR) databases in each of the follow-up surveys [26]. Survival status was noted along with the date of death. It was measured in survival years estimated, starting from 1999 to 2012 using the TNDR record.

#### 2.3.3. Effect Modifier

As we can see, no matter whether elderly with chronic disease or somatic complaint, it will somehow affect the relationship between depression and mortality [9,10,11,12,13,15,16]. Therefore, SRH was determined by asking individuals how they rated their current health. According to the previous research into TLSA, SRH had been carefully evaluated as a useful variable in identifying the frailty group, which may imply a relatively unhealthy physical status [34]. Therefore, we will take it as an effect modifier between depression and mortality in the elderly group during our analysis.

In the TLSA questionnaire, the possible answers were *excellent*, *good*, *fair*, *poor* and *very poor*. In our 3999 subjects, we will divide the individuals into 3 SRH groups: good (rated *excellent* or *good*), fair (*fair*) or poor (*poor* or *very poor*) [34]. We reclassified SRH from five groups to three groups, so that there were enough participants in each group and the result would be easier for further discussion.

#### 2.3.4. Covariates

Age and sex were recorded during the 4th wave of TLSA survey in 1999, and both of these will be used as confounders to calibrate our main study variable. We will take age as a continuous variable. After correcting for these variables, the association between depression and mortality will be reassessed.

### 2.4. Statistical Analysis

A descriptive analysis was performed, calculating the proportion of each subgroup based on mortality and CES-D-10 score in groups. We analyzed the association between depression risk and mortality and other variables by using the Chi square test, student t test, and one-way ANOVA.

A Kaplan–Meier (K-M) curve [35] based on Taiwan’s National Death Registry database linked with our study subjects separated our 3999 subjects into 4 different CES-D-10 groups, and was further stratified by subjects’ SRH. Log-Rank test was used in order to tell the difference in slope among the curves. Survival differences will be evaluated by the multivariate Cox proportional hazards regression model, and the hazard ratio (HR) and 95% confidence interval (CI) were calculated crudely and adjusted for the clinically relevant study covariables (age, sex, and SRH).

In the stratified analysis, we divided our subjects into 3 different SRH groups (good, fair, and poor). The same strategies, K-M curve with L-R test and multivariate Cox proportional hazards regression model will be performed again in 3 different SRH groups to find the differences between the groups. All statistical tests were carried out as two-sided and significant differences were considered at *p*-value < 0.05. Statistical analyses were conducted using STATA version 15 (StataCorp LLC, College Station, TX, USA).

## 3. Result

### 3.1. Demography

There were 3999 subjects included (Table 1). The average age was 66.1, and the sex ratio M:F is 2135:1864. The CES-D-10 score of our data base is 5.4 points, and the case numbers of 4 CES-D-10 groups were 2253, 939, 285 and 522, respectively (Figure 1).

The average CES-D-10 scores of each CES-D-10 group were 1.3, 6.6, 10.9, and 18.1, respectively. The average age rises as the CES-D-10 level grows (65.1, 66.2, 68.3, and 69.1, respectively). In 3 CES-D-10 groups with CES-D-10 score above 5, females were a higher percentage compared with males.

The case numbers of 3 SRH groups were 1434, 1366, and 1199, respectively. As the SRH became worse, the CES-D-10 score rose accordingly.

### 3.2. Characteristic of Mortality

During our 12-year follow up, 1874 of 3999 cases expired, and the mortality prevalence was 46.9% (Table 2). The average age was 8.8 years older in the mortality group, and the mortality rate of males was 10.7% higher than females.

When it comes to CES-D-10 score, the score of the mortality group is 2.4 points higher than the survival group (6.67 and 4.28, respectively). After dividing into four CES-D-10 groups, the mortality prevalence rose as the CES-D-10 level grew (40.7%, 47.82%, 54.39% and 67.62%, respectively).

For SRH, just as in our prediction, the mortality prevalence increased as the SRH went from good to poor (34.4%, 45.3% and 63.3%, respectively).

### 3.3. Relationship between Depression Risk (CES-D-10) and Mortality

The HR showed 1.31 (CI: 1.26–1.36, *p* < 0.001) between CES-D-10 groups before adjusting the covariate, and 1.16 (CI: 1.11–1.22, *p* < 0.001) after adjustment for age, sex, and SRH, which means a 16% increase in instantaneous mortality risk as the CES-D-10 group gets to the next level, and the instantaneous mortality risk of DD group is 56% more than the ND group. The K-M survival estimates of different CES-D-10 score groups are shown as Figure 2, and the L-R test showed *p* < 0.0001. The 4 curves of different CES-D-10 groups are sequentially drifting apart over time.

### 3.4. Stratification by Self-Rated Health (SRH) Status

In the subgroup analysis, we used SRH to stratify the subjects into three groups (Table 3). We can see that, as the SRH gets worse, although the HR is all around 1.1–1.2, the *p*-value becomes smaller and smaller (*p* = 0.122, 0.033, <0.001, respectively), which means the relationship between CES-D-10 and mortality becomes closer as the SRH becomes worse.

When it comes to subgroup K-M survival estimates (Figure 3, Figure 4 and Figure 5), although the L-R test of the 3 groups all showed significance (*p*-value from 0.0235 to <0.0001), from figure to figure, the change of curve representing DP group (5 ≦ CES-D < 10), which is worthy of being noticed. In Figure 3 and Figure 4, in which SRH is good or fair, the two curves representing ND and DP group almost go together, and the gap between DP, MD, and DD group is relatively wide. However, in Figure 5, in which SRH is poor, the gap between ND and DP group becomes wider and the two curves representing DP and MD group almost go together, and even cross.

## 4. Discussion

Depression is a common illness worldwide, with an estimated 3.8% of the population affected, including 5.0% among adults and 5.7% among adults older than 60 years [36]. Depression results from a complex interaction of social, psychological, and biological factors [37].

High levels of depressive symptoms are an independent risk factor for mortality in community-residing older adults. Motivational depletion may be a key underlying mechanism for the depression-mortality effect [38]. Over 12 years of follow-up, a higher CES-D-10 score, which means higher probability of depression, is related to higher risk of mortality, and the result is consistent with previous studies [9,10,11,12,13,15,16].

According to the result, the percentage of females was higher in DP, MD, and DD group, which means females are more likely to develop mood disorder and this is consistent with past studies [39]. Although minimal, the average age got higher as CES-D-10 group went from ND to DD, which that hints depression prevalence might increase with age [2]. Some mechanisms have already been identified in a systemic review showing that depression predisposes to medical illnesses and advances biological aging, which may be caused by shorter telomere length, accelerated brain aging and advanced epigenetic aging [40].

Unsurprisingly, worse SRH, which can represent poor physical condition [34], is also related to higher mortality risk. However, due to its self-rated nature, the limitation is that we cannot infer the precise meaning of the individuals’ SRH result. For example, some of the elderly may report their true physical condition, while others may report according to their recent experience (or the view of physician, friend, or family) about whether they met the standard of healthy.

There are interrelationships between depression and physical health. For example, cardiovascular disease can lead to depression and vice versa. In our study, a higher CES-D-10 score was noted in the worse SRH group, which also compared with a previous study [17,41]. Debate between the causal effect of physical and psychiatric problem never stop [4,5]. However, in our cohort study, due to the limitation of a retrospective longitudinal nature, the data acquisition of CES-D-10 and SRH took place at the same survey time, so it is difficult to tell the causal effect between these two variables. However, we can see co-creation of the two causes, contributing to a clearer picture of mortality risk.

We used to use the GDS or CES-D for screening and assessment of late-life depression, and we are now used to using a uniform standard for all geriatric populations, but in the process of clinical intervention, we understand that each individual is an individual, with a different health status. In the study of the UK system, education is taken as an important factor in predicting long term care and health of the elderly; in a study of the American system, SRH is taken as a good predicting factor for physical and psychological health in the elderly. For a clearer and more generalizable definition, in this study we choose SRH to be our main effect modifier in order to discuss the relationship between depression and mortality.

After the stratification of SRH, we can see some distinctive findings. First, after setting SRH as an effect modifier, the relationship between mortality and depression became stronger as individuals’ SRH became worse, *p* value from 0.122 to <0.001, and the increase in HR is also obvious from 1.1 to 1.2. Next, in analyzing the trend for Figure 3 or Figure 4, the four curves went almost together in the first 5 years, which means limited difference in 5-year-mortality, but in the long run we can see the curve for ND and DP going almost together, which made 10 points a perfect cutting point for SRH showing good and fair. However, when it comes to the trend for Figure 5, the curve for DP group (CES-D-10: 5~9) went together with the curve for MD group (CES-D-10: 10~12), even crossing at the end point of our study, which made 5 points a more suitable cutting point for SRH showing poor. All the results point to the same idea, that in individuals with worse SRH, the relationship between mortality and depression became stronger, and any small change in CES-D-10 score will more likely lead to an increase in mortality. This somehow implies that we should set a relatively lower cutting point of CES-D-10 for depression diagnosis in worse SRH individuals in order to achieve long-term health goals.

Depression is not a normal part of the aging process. Depression in older adults is a treatable medical condition; a variety of psychotherapeutic options are available. Older patients must be viewed in their medical, functional, and social context for effective management [42]. We preferred to set the tone of our research as a pilot study to bring out the idea that not only can depression alone contribute to a rise in mortality, but with a relatively poor physical condition its impact on mortality will also be more pronounced. Future research may consider taking more physical health details and socioeconomic status into consideration, in order to create the whole picture about how physical and psychological factors interact. When conducting research in the geriatric population, we should consider the cutting point of the depression screen tool very carefully according to physical health status. For relatively high risk individuals, prevention programs such as physical exercise, social support [42] and social participation [33], which have been shown to be effective [43], should be given as early as possible [36].

## 5. Conclusions

We conclude that depression is an important impact factor in long-term health in the elderly. Mortality as a final result of health care is enormously affected by emotion. The importance of the relationship between depression and mortality is re-emphasized in our study.

Moreover, through adding SRH in our analysis, we can conclude that in SRH poor any sign of depression may lead to a rise in mortality, which means depressive symptoms may have a more profound effect on health in individuals with SRH showing as poor. This result meets our hypothesis that co-creation of the two causes contributes to a clearer picture of mortality risk, which may represent the long-term prognosis. So, when evaluating the elderly, we must pay attention to their psychological status, and remember that depressive mood should be scrutinized more carefully in the elderly who feel themselves unhealthy and a preventive program should be offered as soon as possible.

## Figures and Tables

**Figure 1 ijerph-19-06922-f001:**
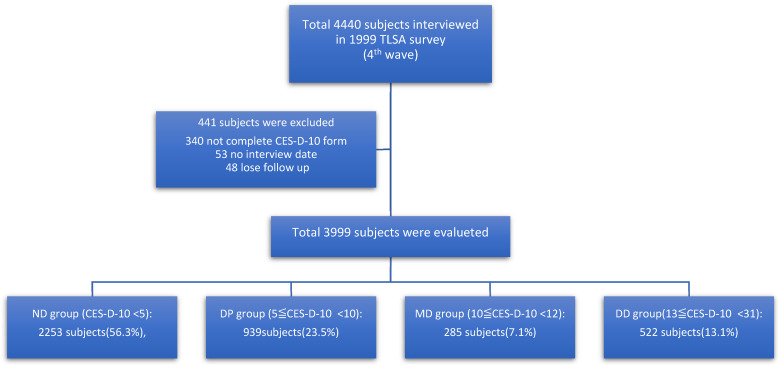
A flow diagram of the subject-inclusion process.

**Figure 2 ijerph-19-06922-f002:**
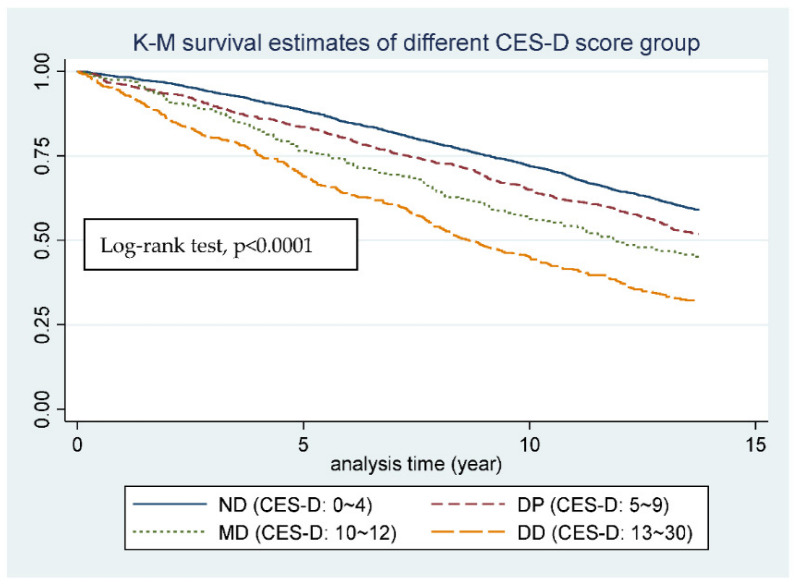
K-M survival estimates of different CES-D score groups.

**Figure 3 ijerph-19-06922-f003:**
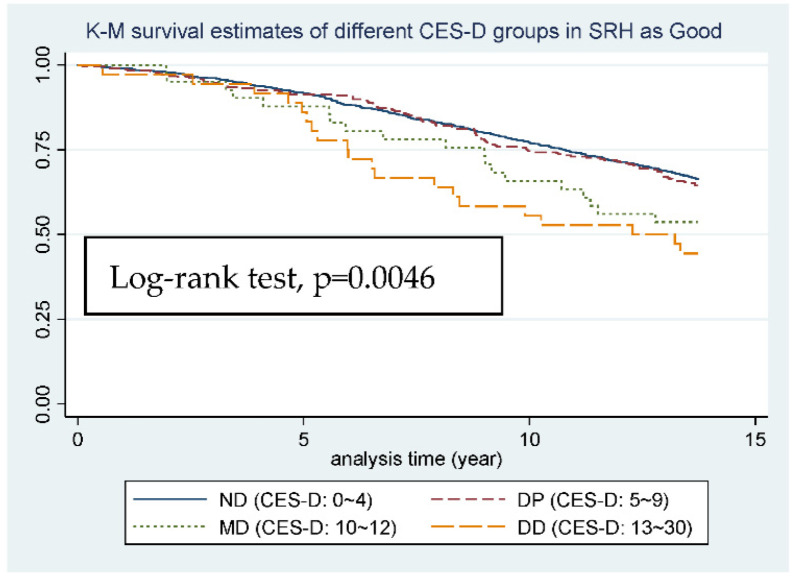
K-M survival estimates of different CES-D groups in SRH as Good.

**Figure 4 ijerph-19-06922-f004:**
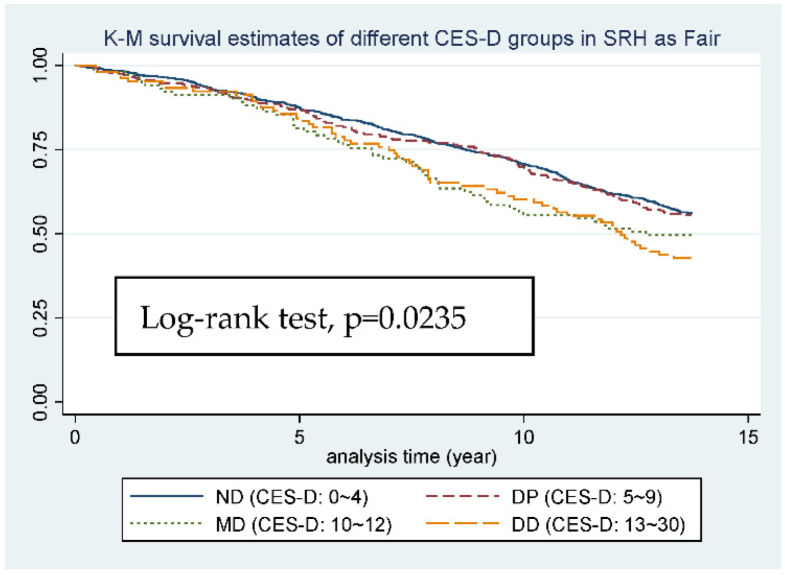
K-M survival estimates of different CES-D groups in SRH as Fair.

**Figure 5 ijerph-19-06922-f005:**
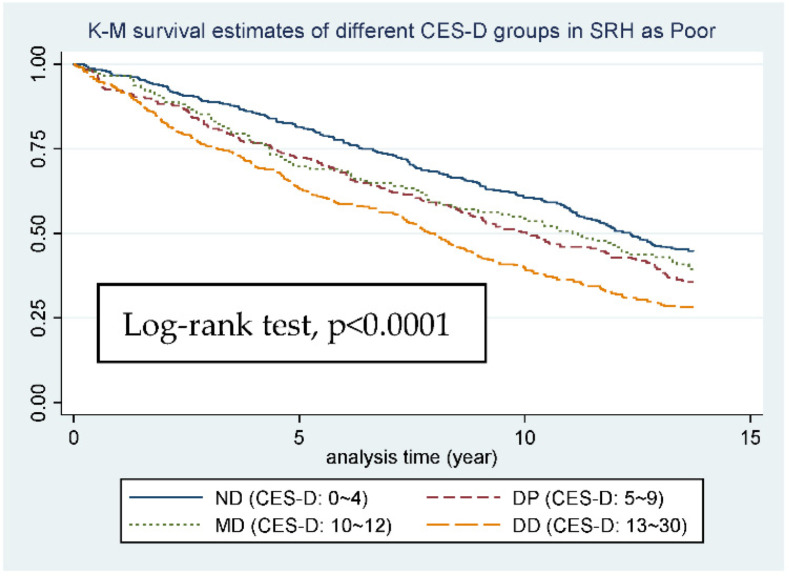
K-M survival estimates of different CES-D groups in SRH as Poor.

**Table 1 ijerph-19-06922-t001:** Characteristic by Different CES-D Score Groups.

CES-D in Group		Total	No Depression (ND)		Depression Potential (DP)		Mild Depression (MD)		Depression Disorder (DD)			*p*-Value
	CES-D Score		CES-D: 0~4		CES-D: 5~9		CES-D: 10~12		CES-D: 13~30			
Number of subjects		3999	2253	56.34%	939	23.48%	285	7.13%	522	13.05%		
Average CES-D-10 score		5.4	1.26		6.63		10.87		18.08			
Average Age		66.1	65.07		66.17		68.28		69.12		F = 36.85	<0.0001
Sex	Male	2135	1350	63.23%	458	21.45%	113	5.29%	214	10.02%	chi^2^ = 100.49	<0.001
	Female	1864	903	48.44%	481	25.80%	172	9.23%	308	16.52%		
Self-rated Health (SRH)	Good	1434	1049	73.15%	308	21.48%	41	2.86%	36	2.51%	chi^2^ = 775.95	<0.001
	Fair	1366	837	61.27%	323	23.65%	102	7.47%	104	7.61%		
	Poor	1199	367	30.61%	308	25.69%	142	11.84%	382	31.86%		

**Table 2 ijerph-19-06922-t002:** Characteristics by Mortality.

		Total	Alive	Death	Mortality Rate		*p*-Value
			0	1	%		
Number of subjects		3999	2125	1874	46.86%		
Average Age		66.1	62.2	71		t = −33.44	*p* < 0.0001
Sex	Male	2135	1028	1107	51.85%	chi^2^ = 45.77	<0.001
	Female	1864	1097	767	41.15%		
Average CES-D-10 score		5.4	4.28	6.67		t = −12.66	*p* < 0.0001
CES-D in group	No depression (ND)	2253	1336	917	40.70%	chi^2^ = 131.53	<0.001
	Depression potential (DP)	939	490	449	47.82%		
	Mild depression (MD)	285	130	155	54.39%		
	Depression disorder (DD)	522	169	353	67.62%		
Self-Rated Health (SRH)	Good	1434	938	496	34.59%	chi^2^ = 218.21	<0.001
	Fair	1366	747	619	45.31%		
	Poor	1199	440	759	63.30%		

**Table 3 ijerph-19-06922-t003:** Hazard Ratio in change of every CES-D Score in Group, stratified by SRH Condition.

Mortality Risk of 4 CES-D-10 Groups		Number of Subjects	Hazard Ratio	95% CI		Z	*p*-Value
Adjusted with Age/Sex/SRH		3999	1.16 ***	1.11	1.22	6.81	<0.001
Stratification by Self-Rated Health (SRH)	Good	1434	1.10	0.97	1.24	1.55	0.122
	Fair	1366	1.09 *	1.01	1.19	2.32	0.033
	Poor	1199	1.20 ***	1.13	1.27	6.16	<0.001

* *p* value < 0.05; *** *p* value < 0.001.

## Data Availability

The datasets generated during the current study are not publicly available, but data are however available from the applicants upon reasonable request and with permission of the Ministry of Health and Welfare in Taiwan.

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
