# Peer review of "The Influence of Depressive Mood on Mortality in Elderly with Different Health Status: Evidence from the Taiwan Longitudinal Study on Aging (TLSA)"

_ijerph, 2022, doi:10.3390/ijerph19116922_

Round 1

Reviewer 1 Report

This article has the following minor/major drawbacks:

1.      Please rewrite the aim of the study in the Abstract. I suggest including the two objectives of the study in the Abstract.

2.      Please separate the related sections to Methodology from Introduction (Page 1, lines 46-58).

3.      There is a lack of theoretical support concerning the conceptual framework of this study. Especially it is not clear how the second objective of this study is supported empirically/theoretically by the previous studies.

4.      I could not understand well that based on what criteria, CES-D-10 was divided into 4 groups.

5.      The section of discussion needs to be extended by reviewing the relevant studies.

6.      It seems that the section on Conclusion also needs to be extended.

7.      The limitations of this study especially regarding the method are not clear.

Author Response

Dear Professor Reviewer:

Thanks so much for your professional review and kind comments on our manuscript. We have carefully checked and revised ( in yellow color in the text) accordingly to the comments:

  1. Please rewrite the aim of the study in the Abstract. I suggest including the two objectives of the study in the Abstract.
    ANS: Dear Professor Reviewer, thanks so much for your reminding. We have added our 2 objectives into the abstract. (Page 1, lines 17-21, Page 2 , lines 28-30)

  2. Please separate the related sections to Methodology from Introduction (Page 1, lines 46-58).
    ANS: Dear Professor Reviewer, we have separated the related sections about CES-D-10 from Introduction to Methodology, thank you for your reminding!

  3. There is a lack of theoretical support concerning the conceptual framework of this study. Especially it is not clear how the second objective of this study is supported empirically/theoretically by the previous studies.
    ANS: Dear Professor Reviewer, thanks so much for your reminding. We have added in some sentence to improve the continuity of our setting from depression to mortality and the role of self-rated health, make the whole framework much strong. (Page 2, lines 15-17, 25, 26)

  4. I could not understand well that based on what criteria, CES-D-10 was divided into 4 groups.
    ANS: Dear Professor Reviewer, thanks so much for your reminding. We have re-emphasized the reason why we set 4 cut point in the introduction and method to make the idea more clear. (Page 3, lines 20-22, 27, 28, 31, 32)

  5. The section of discussion needs to be extended by reviewing the relevant studies.
    ANS: Dear Professor Reviewer, thanks so much for your concerns. We have added on few more review article(Reference 36-43: about gender & depression, late-life depression mechanism, and systemic review about depression & frailty) in order to support our finding in this study. (Page 9-10)

  6. It seems that the section on Conclusion also needs to be extended.
    ANS: Dear Professor Reviewer, thanks so much for your concerns. We have done some extend from the material in our discussion. We are trying our best to make the conclusion look short and clear to bring out the most important idea that the relationship between depression and mortality is more strong in relatively unhealthy elderly. (Page 9, conclusion)

  7. The limitations of this study especially regarding the method are not clear.
    ANS: Dear Professor Reviewer, thanks so much for your concerns. We have emphasized some points of limitation in our discussion which included the cohort setting and self-rated nature. Thank you for your reminding. (Page 9, lines 18,19,27,28)

Dear Professor Reviewer,

Thanks again for your concerns and comments. We do have carefully checked and revised it as we can. Hopefully this manuscript could be better and much fruitful to our colleagues.

Reviewer 2 Report

This is an epidemiological study on the association between depression and mortality From the Taiwan Longitudinal Study on Aging (TLSA). The study benefits from the linkage of survey data to a wide variety of other data sources including information on health and social care.

General comments: Highly suggested for English proof-reading.

I spotted “a face-to-face nterview” (line 88) in which an I is missing, and Future research may consider taking more physicalogical health detail and socioeco-….  (line 64)

My suggestions for further improvements are as follows:

  1. In the methodology, I think it would be helpful to clarify that this protocol is, in effect, for a full longitudinal study.
  2. A justification of the sample size (4049 in 1989 or 4440 in 1999) needs to be better stated.
  3. The sampling description is highly technical and no citation is provided for the original source for this method. The authors allude to some problems with this sampling plan but neither elaborate nor justify their choice, particularly in using the 4th wave data.
  4. Main Computer-Assisted Personal Interviews (CAPI) and self-completion questionnaires - Are these questionnaires ready to be shared as appendices?
  5. To discuss the biological mechanisms of the depression in elderly. Depression predisposes to a variety of medical illnesses, but also medical illnesses increase the risk of late-life depression. Late-life and mid-life is often associated with medical and psychosocial problems at the individual as well as at the community level. This leads to acceleration of inflammatory responses, increased oxidative stress, alteration in neurogenesis and prefrontal cortex network and functional connectivity.
  6. To discuss the limitations and future directions of the study.

Author Response

Dear Professor Reviewer,

Thanks so much for your professional review and kind comments on our manuscript. We have carefully checked and revised ( in yellow color in the text) accordingly to the comments:

I spotted “a face-to-face nterview” (line 88) in which an I is missing, and Future research may consider taking more physicalogical health detail and socioeco-…. (line 64)
ANS: Dear Professor Reviewer, thanks so much for your concerns. We have gone through the article once again to prevent such mistake.

For further improvements are as follows:

  1. In the methodology, I think it would be helpful to clarify that this protocol is, in effect, for a full longitudinal study.
    ANS: Dear Professor Reviewer, thanks so much for your reminding. We have re-emphasized this part in our methodology. (Page 2, lines 38, 42, 43, Page 2, lines 1)

  2. A justification of the sample size (4049 in 1989 or 4440 in 1999) needs to be better stated.
    ANS: Dear Professor Reviewer, thanks so much for your reminding. We have strengthened this part of instructions this part in our methodology. (Page 3, lines 3-5)

  3. The sampling description is highly technical and no citation is provided for the original source for this method. The authors allude to some problems with this sampling plan but neither elaborate nor justify their choice, particularly in using the 4th wave data.
    ANS: Dear Professor Reviewer, thanks so much for your concerns. In our study setting, the last data we can acquire from Taiwan National Death Registry (TNDR) is until 2012/12/31. In order to obtain more than 10 years observation, which can give the outcome more meaningful explanation, we choose 1999 as our origin in order to meet the object.

  4. Main Computer-Assisted Personal Interviews (CAPI) and self-completion questionnaires - Are these questionnaires ready to be shared as appendices?
    ANS: Dear Professor Reviewer, thanks so much for your concerns. We will give the scanning of the cover of our questionnaire (the one we used is whitened in Chinese) , but since the study started long time ago, we don’t have the formal English official file, so we will have some difficulty in sharing it as our appendices.

  5. To discuss the biological mechanisms of the depression in elderly. Depression predisposes to a variety of medical illnesses, but also medical illnesses increase the risk of late-life depression. Late-life and mid-life is often associated with medical and psychosocial problems at the individual as well as at the community level. This leads to acceleration of inflammatory responses, increased oxidative stress, alteration in neurogenesis and prefrontal cortex network and functional connectivity.
    ANS: Dear Professor Reviewer, thanks so much for your concerns. We have added on some reference about mechanism of late-life depression in our discussion. (Page 9, lines 14-16)

  6. To discuss the limitations and future directions of the study.
    ANS: Dear Professor Reviewer, thanks so much for your concerns. We have emphasized some points of limitation in our discussion which included the cohort setting and self-rated nature. Thank you for your reminding. (Page 9, lines 18,19,27,28)

Dear Professor Reviewer,

Thanks again for your concerns and comments. We do have carefully checked and revised it as we can. Hopefully this manuscript could be better and much fruitful to our colleagues.

Reviewer 3 Report

This is a report on depression and mortality in a large sample of Taiwanese.
Please clarify regarding the following.

It seems to me that you are presenting slightly outdated data. Do you have more recent data?

How do you know that there are no identical patients among the 4400 people included in the study?

What is the life expectancy of Taiwanese people? It seems to be a group with a somewhat shorter than average life expectancy.

Author Response

Dear Professor Reviewer,

Thanks so much for your professional review and kind comments on our manuscript. We have carefully checked and revised ( in yellow color in the text) accordingly to the comments:

  1. It seems to me that you are presenting slightly outdated data. Do you have more recent data?
    ANS: Dear Professor Reviewer, thanks so much for your concerns. In our study setting, it’s a pity that the last data we can acquire from Taiwan National Death Registry (TNDR) is until 2012/12/31. In order to obtain more than 10 years observation, which can give the outcome more meaningful explanation, we choose 1999 as our origin in order to meet the object. Thankfully, our study object is to clarify the relationship between depression, physical health and mortality, so there won’t be too much change as time goes by as other high-tech related study. However, TLSA data is still on going, we can expect that more comprehensive analysis of the elderly will be launched in succession.

  2. How do you know that there are no identical patients among the 4400 people included in the study?
    ANS: Dear Professor Reviewer, thanks so much for your concerns. All the participant will have a certain number of identification, and can receive the survey only once in each wave, so the possibility to have identical participants is small.

  3. What is the life expectancy of Taiwanese people? It seems to be a group with a somewhat shorter than average life expectancy.
    ANS: Dear Professor Reviewer: The average lifespan of Taiwanese is 79.2-year-old in Aug, 2012. (Male 76.1-year-old; Female 82.63-year-old) The average age of our study subjects is 66.1-year-old in 1999 as we mentioned in the manuscript. So, after 12 years observation (2012), the average age of our subject will be 78.1 years old, and 46.86% had expired. The average of life expectancy of our study subjects is not too far away from lifespan in Aug, 2012. Thanks so much for your concerns.

Dear Professor Reviewer,

Thanks again for your concerns and comments. We do have carefully checked and revised it as we can. Hopefully this manuscript could be better and much fruitful to our colleagues.

Round 2

Reviewer 1 Report

Thanks to the authors, I do not have any more comments.

Reviewer 3 Report

The manuscript has been much improved.